# Current Knowledge on Graves’ Orbitopathy

**DOI:** 10.3390/jcm10010016

**Published:** 2020-12-23

**Authors:** Katarzyna Gontarz-Nowak, Magdalena Szychlińska, Wojciech Matuszewski, Magdalena Stefanowicz-Rutkowska, Elżbieta Bandurska-Stankiewicz

**Affiliations:** Clinic of Endocrinology, Diabetology and Internal Medicine, School of Medicine, Collegium Medicum, University of Warmia and Mazury in Olsztyn, 10-957 Olsztyn, Poland; szychlinskam@gmail.com (M.S.); wmatuszewski82@wp.pl (W.M.); m.m.stefanowicz@gmail.com (M.S.-R.); bandurska.endo@gmail.com (E.B.-S.)

**Keywords:** Graves’ orbitopathy, Graves’ disease, clinical activity score, immunosuppressive treatment, glucocorticoids, radiotherapy

## Abstract

(1) Background: Graves’ orbitopathy (GO) is an autoimmune inflammation of the orbital tissues and the most common extra-thyroid symptom of Graves’ disease (GD). Mild cases of GO are often misdiagnosed, which prolongs the diagnostic and therapeutic process, leading to exacerbation of the disease. A severe course of GO may cause permanent vision loss. (2) Methods: The article presents an analysis of GO—its etiopathogenesis, diagnostics, current treatment and potential future therapeutic options based on a review of the currently available literature of the subject. (3) Results: Current treatment of the active GO consists predominantly in intravenous glucocorticoids (GCs) administration in combination with orbital radiotherapy. The growing knowledge on the pathogenesis of the disease has contributed to multiple trials of the use of immunosuppressive drugs and monoclonal antibodies which may be potentially effective in the treatment of GO. Immunosuppressive treatment is not effective in patients in whom a chronic inflammatory process has caused fibrous changes in the orbits. In such cases surgical treatment is performed—including orbital decompression, adipose tissue removal, oculomotor muscle surgery, eyelid alignment and blepharoplasty. (4) Conclusions: Management of GO is difficult and requires interdisciplinary cooperation in endocrinology; ophthalmology, radiation oncology and surgery. The possibilities of undertaking a reliable assessment and comparison of the efficacy and safety of the therapeutic strategies are limited due to the heterogeneity of the available studies conducted mostly on small group of patients, with no comparison with classic systemic steroid therapy. The registration by FDA of Teprotumumab, an IGF1-R antagonist, in January 2020 may be a milestone in future management of active GO. However, many clinical questions require to be investigated first.

## 1. Introduction

The Graves’ orbitopathy (GO) is an autoimmune inflammation of the orbital tissues and the most common extra-thyroid symptom of Graves’ disease (GD). GO occurs in 25–50% of patients with GD, although the literature shows that subclinical ocular lesions can be observed in the majority of patients with GD when high-quality imaging techniques are used [1,2,3,4,5]. Mild cases of GO are often misdiagnosed as conjunctivitis or allergic symptoms, which prolongs the diagnostic and therapeutic process, in some cases leading to exacerbation of the disease [6]. Symptoms typical of GO—ocular pain, excessive tearing, photophobia, visual disturbances, including double vision—significantly reduce the quality of life of patients [7]. The inflammatory state in the orbit manifests itself mainly as redness and swelling of conjunctivas and eyelids, exophthalmos, and retrobulbar pain. A severe course of GO with the occurrence of dysthyroid optic neuropathy (DON) or corneal ulceration may lead to permanent vision loss [8]. The current treatment of active GO consists predominantly in intravenous administration of glucocorticoids (GCs); however, according to the literature, the results of such treatment are unsatisfactory in 35% of cases. Immunosuppressive treatment is not effective in patients in whom a chronic inflammatory process has caused fibrous changes in the orbits [9]. For this reason, early identification of patients at risk of severe GO is of highest significance, since effective treatment of hyperthyroidism, careful observation of ocular lesions and prompt implementation of appropriate treatment can significantly alleviate the course of the disease.

## 2. Etiopathogenesis

The onset of GO is usually closely related to GD in its hyperthyroidism phase. Orbitopathy most often develops synchronously, although it can also precede or follow the occurrence of hyperthyroidism [10]. Thyrotropin receptor antibodies (TRAbs), pathognomonic for GD, are present in every patient with GO, and their concentration positively correlates with the severity and the activity of the disease [11]. The thyrotropin receptor (TSH-R), present on the thyroid cells, is also physiologically found on the surface of orbital fibroblasts, yet in the case of GO, it is overexpressed [12,13,14].

After binding to the TSH-R on orbital fibroblasts, TRAbs activate the immune cascade leading to the infiltration of the activated B and T lymphocytes, as well as bone marrow-derived CD34 + fibrocytes, which differentiate into myofibroblasts or adipocytes. The incoming cells release numerous cytokines and chemokines, such as interferon gamma (IFN-y), tumour necrosis factor alpha (TNF-α), interleukin-1β (IL-1β), interleukin-2 (IL-2), interleukin-6 (IL-6) as well as leukoregulin, which strongly stimulate the local synthesis of glycosaminoglycans (GAG), including hyaluronic acid (HA). Accumulation of the strongly hydrophilic HA causes local water retention and swelling of connective tissue and extraocular muscles, which in turn worsens the venous and lymphatic circulation in the orbit. The activation of periocular fibroblasts, which are known to be progenitor fat cells, leads to the enlargement of orbital fat tissue. As a result, the intra-orbital pressure increases leading to subsequent protrusion of the eyeballs forward beyond the edge of the orbit, what clinically manifests as proptosis or exophthalmos. The inflammatory process of the oculomotor muscles impairs their function and disables the coordinated movement of the eyeballs causing double vision. A long-term inflammatory state leads to gradual muscle remodeling and fibrosis, which results in persistent mobility disorders of the eyeballs [14,15,16]. According to some authors, the autoimmune process of GO, besides aforementioned cytokines and chemokines, involves also some growth factors [17]. Indeed, overexpression of the insulin-like growth factor 1 receptor (IGF1-R) is observed on orbital fibroblasts, as well as on the infiltrating T and B lymphocytes. Studies suggest that IGF1-R is involved in the signaling pathway induced by TRAb ligation with TSH-R, through which it is transactivated. Being part of the aforementioned complex, IGF1-R enhances the TSHR-dependent production of cytokines, leading to increased release of inflammatory mediators and, as a result, increased production of HA. The initial hypothesis about the presence of specific anti-IGF1-R autoantibodies involved in the pathogenesis of GO has not been confirmed [18,19,20,21,22]. The aforementioned signaling cascade is presented on Figure 1.

## 3. Risk Factors

The onset of GO appears to be conditioned by a complex interaction of genetic and environmental factors. According to the literature some predisposing alleles for GD development have been determined. Moreover, some genetic divergences between GD patients with and without orbitopathy have been recognized, however none of the polymorphisms have proved adequately predictive to support genetic testing in determining prevention methods and further diagnostic and therapeutic process [23,24,25]. Multiple studies revealed predominance of GO in women, with a 2:1, ratio. [26,27]. On the other hand, more severe forms of GO are observed in men [26]. It is noteworthy that ethnicity also has a significant impact on GO occurrence in GD patients—it was proved that Europeans are at more than six times higher risk of GO than Asians and Indians. The predominant manifestations of GO also vary among the ethnic groups. In Caucasians, soft tissue involvement and retraction of the upper eyelid are the most frequent symptoms, whereas in Asians exophthalmos and lower lid retraction are more common [25,28,29]. There is a complete gap in the literature about prevalence and clinical manifestations of GO in African population. Furthermore, the age of the patient and the duration of GD-related hyperthyroidism correlate positively with the risk of GO development [30]. Moreover, older age of onset of GO is associated with more severe course of disease [26]. 

High serum TRAb titer increases the risk of GO development, positively correlate with the activity and severity of the disease and are a predictor of poor response to the to immunosuppressive treatment and the risk of relapse after treatment [31].

The prevalence of type 1 diabetes mellitus (DM) in patients with GO is higher than in the normal population. Moreover, DON occurs more prevalently in patients with GO and DM than in the total group of GO patients, thus DM seems to be a risk factor of more severe course of the disease [32]. 

Many exogenous factors also influence the occurrence and course of GO. First, cigarette smoking, possibly through its impact on both humoral and cellular immunity, is one of the strongest risk factors [14,24]. Smoking, both active and passive, is associated not only with the higher risk of de novo development of GO, but also with more severe eye symptoms and delayed and limited outcomes of the immunosuppressive therapy [33,34,35]. In addition, longer persistence and higher serum TRAb titer during and after the treatment of GO have been observed in smokers, which may be responsible for a higher incidence of steroid resistance and steroid dependence in this group of patients [31]. A higher de novo occurrence or exacerbation of pre-existing GO has also been reported after radioiodine treatment in tobacco smokers [36,37].

Second, transient thyroid dysfunction, both hyperthyroidism and hypothyroidism, is associated with a greater risk of development, progression, and severe course of orbitopathy compared to euthyroid patients [38]. Third, radical treatment of hyperthyroidism with radioiodine (131-I), possibly through an increase in the serum TRAb concentration caused by transient inflammation of the thyroid gland, may cause de novo development of GO and significantly increases the risk of progression of an already existing disease [5,31,38].

The aforementioned risk factors are presented in Table 1.

## 4. Clinical Picture and Diagnosis

The onset of GO appears to be conditioned by a complex interaction of genetic and environmental factors. According to the literature some predisposing alleles for GD development have been determined. Moreover, some genetic divergences between GD patients with and without orbitopathy

The diagnosis of GO is based mainly on the typical ocular symptoms in patients with Graves’ hyperthyroidism. It should be emphasized that ocular lesions may also precede the occurrence of hyperthyroidism, occur without it (euthyroid Graves’ orbitopathy) or accompany hypothyroidism both during the treatment of GD and in the course of Hashimoto’s thyroiditis [39].

The inflammatory process is manifested by: redness and swelling of the conjunctivas, eyelids and the lacrimal caruncle; a sensation of distension in the orbit, pain behind the eyeballs; and impaired mobility of the oculomotor muscles. Retraction of the upper eyelid and exophthalmos lead to the exposure of the cornea what results in its irritation, which is described by patients as a sensation of a foreign body under the eyelids, and often leads to compensatory excessive tearing [7]. Clinical picture of GO has been shown in Figure 2. The course of GO is usually bi-phasic—after a period of active inflammation lasting from 18 to 36 months, there is a chronic inactive phase [40]. Among laboratory tests which are instrumental in providing a diagnosis, the TRAb antibody titer is of the greatest value.

We can see swelling and redness of the upper and lower eyelids, redness of the conjunctivae. The eyeballs located centrally. Binocular restriction of the upwards motility of the eyeballs, in other directions correct (not shown on the picture). Exophthalmos measurements within the normal range. CAS 5/7.

## 5. Clinical Classification

Therapeutic management depends on the severity of orbitopathy as well as the activity of the inflammatory process. In 1989, a clinical classification called the Clinical Activity Score (CAS) was proposed, the aim of which was to easily distinguish between the active and stationary phase of the disease, referring to the classic symptoms of acute inflammation, such as: pain, redness, swelling and functional impairment. Until today, a modified version of the CAS scale has been used. One point is given for the presence of each of the seven assessed symptoms as presented in Table 2. A score equal or greater than three points indicates an active inflammatory process and the potential effectiveness of immunosuppressive therapy [9].

As mentioned above, it is also important in clinical practice to assess the severity of the disease with the aim to identify patients with the highest risk of sight-threatening course of GO. Classifications used for this purpose—NO SPECS (No signs or symptoms; Only signs or symptoms; Soft tissue involvement; Proptosis; Extraocular muscle involvement; Corneal Involvement; Sight loss) [41] and VISA (Vision, Inflammation, Strabismus and Appearance) [42]—are shown in Table 3 and Table 4, respectively. 

The European Group on Graves’ Orbitopathy (EUGOGO) proposed a classification that enables the assessment of both the activity and the severity of the disease [40]. Based on a detailed analysis of the eye symptoms from each category with the use of measurable parameters (according to the scheme described in Table 5 and the atlas available on the EUGOGO website), orbitopathy is classified as mild, moderate-to-severe and sight-threatening, respectively, and further therapeutic decisions are based on this classification, as shown in Table 6.

## 6. Image Evaluation

The diagnosis of GO is mostly based on clinical signs and symptoms. However, imaging studies facilitate proper diagnosis, differentiation between the active inflammatory phase and the fibrotic end stage, planning surgical orbital decompression and the follow-up assessment after treatment [43]. It should be emphasized that imaging tests should always be conducted in cases of ambiguous clinical features to perform a differential diagnosis with other orbital pathologies. Moreover, image evaluation enables identification of patients at high risk of DON development in whom early introduction of treatment can prevent vision loss [43]. Currently, magnetic resonance imaging (MRI) is the imaging method of choice in GO enabling a precise orbital soft-tissue evaluation, and thus, assessment of the activity of the inflammatory process [39,44]. Typical for GO MRI findings include proptosis, enlargement of intra- and extra-conal fat and extraocular muscles and, in severe cases, compression of the optic nerve in the orbital apex, known as crowded apex syndrome [45]. The standardized protocol includes coronal fast spin echo T1 and strong T2-weighted sequences called TIRM (turbo-inversion recovery magnitude) [44,46]. Transverse MRI images of patients with GO have been shown on Figure 3 and Figure 4.

Though not accurate in defining the activity of the orbitopathy and the edema of soft tissues and muscles, computed tomography (CT) is a modality of choice due to its precise imaging of bone structures before planning orbital decompression in the inactive phase of GO, to navigate during the surgery and as a follow-up assessment after decompression. However, despite the short duration of the examination and precise evaluation of bone structures, the use of CT is limited also due to the radiation exposure [47].

Numerous studies have also assessed the use of ultrasound in the imaging of the orbit in GO due to its low costs, short time of investigation and no risk of radiation. However, ultrasound is a very observer-dependent modality. Compared to MRI, it is not sufficiently precise in evaluating edema of the extraocular muscles and orbital fat tissues, thus, it does not provide a proper information about the activity of GO. Moreover, insufficient imaging of bone structures, when compared to CT, is another limitation of this modality [48]. Color doppler imaging (CDI), beside conventional grayscale ultrasound image, allows the visualization of blood flow direction and velocity in real time [49]. CDI can be a valuable modality in GO due to the fact that active orbital inflammation is characterized by increased orbital blood flow. Moreover, it was supposed that venous congestion is one of the components of the pathogenesis of GO and its assessment before surgical treatment and as a follow-up imaging may prove valuable [50].

Due to the expression of somatostatin receptors on activated T-lymphocytes, 111In-labeled octreotide scintigraphy was proposed as an alternative diagnostic tool in GO characterized by a high sensitivity in identifying the inflammatory process of orbital tissues [51]. Positive octreotide intake correlates with the activity of orbital inflammation and is a predictor of the effectiveness of immunosuppressive treatment [45]. Nevertheless, this modality is not useful in inactive GO and does not provide any information about the morphology or anatomy of the imaged area. High costs and exposure to radiation are other disadvantages of this technique, which significantly limit the use of octreoscan as a routine imaging procedure in GO.

Multiple studies revealed a positive relation between the activity of GO and diethylenetriamine pentaacetic acid (DTPA) orbital uptake. Furthermore, in the literature there are studies that assessed the use of Technetium-99m labeled DTPA single photon emission tomography (99mTc-DTPA SPECT/CT) to evaluate the inflammatory process in the extraocular muscles of GO patients and to predict the response of immunosuppressive treatment [52]. Increased 99mTc–DTPA uptake is associated with the degree of vascularization and inflammation, thus, correlating positively with the activity of GO. Moreover, it was proved that positive 99mTc–DTPA SPECT/CT can indicate patients, who despite of having low CAS could benefit from immunotherapy. The possible explanation of this inconsistency may be that CAS evaluates eye symptoms and is a very observer-dependent classification, whereas 99mTc–DTPA SPECT/CT assesses the actual orbital inflammatory state. 99mTc–DTPA SPECT/CT can also prove valuable in the follow up assessment after treatment [52].

## 7. Management of Graves’ Orbitopathy

### 7.1. Modifiable Risk Factors

In view of the aforementioned considerations about risk factors in GO, smoking cessation appears to be crucial in primary, secondary and tertiary prevention of GO. Therefore, all GD and GO patients should be advised by their physicians to quit smoking and receive the necessary specialistic support [53]. Furthermore, early diagnosis and selection of the optimal therapy for hyperthyroidism can significantly improve the course of orbitopathy. As radioiodine treatment increases the risk of de novo development or progression of the pre-existing orbitopathy it is recommend that patients with active mild GO receive prophylactic oral glucocorticoid therapy (0.3–0.5 mg of prednisone per one kilogram of body mass per day with gradual dose reduction over 3 months) during the 131-I treatment [54,55]. Prophylactic use of oral steroid therapy in case of 131-I treatment in patients without pre-existing orbitopathy remains controversial. It needs emphasizing that active GO classified as moderate-to-severe and sight-threatening is a contraindication to 131-I treatment of hyperthyroidism. In such cases pharmacological treatment with anti-thyroid drugs is administered or thyroidectomy is performed, which according to the literature have no significant effect on the course of GO [54,56]. Regardless of the treatment method, regular assessment of serum thyroid hormone concentration and prompt restoration of euthyroidism in case of thyroid dysfunction are of major significance as transient thyroid dysfunction, both hyperthyroidism and hypothyroidism, is associated with a greater risk of development, progression, and severe course of GO [57].

### 7.2. General Principles of GO Management

The management of GO depends on the disease severity and activity, as presented in Figure 5. However, the guidelines emphasize the importance of individualizing the treatment strategy, so that potential benefits do not outweigh the possible side effects. It is recommended in routine clinical practice that the quality of life of the patients is estimated based on the use of a validated GO-specific quality of life tool (GOQoL), which can be easily accessed on the EUGOGO website [53]. As management of GO requires interdisciplinary cooperation in endocrinology, ophthalmology, radiation oncology and surgery, patients with GO should be referred by general practitioners and internal medicine specialists to specialized centers, except for the mildest courses of the disease [53].

All GO patients, regardless of the disease severity, should have the following measures implemented [1]:-Artificial tears containing osmoprotective agents, such as sodium hyaluronate, with moisturizing eye drops should be applied regularly to alleviate symptoms of corneal irritation;-Gels or ointments may be required in cases of significant corneal exposure, mainly at night;-Protective glasses with an ultraviolet (UV) filter;-Anti-inflammatory and antibacterial ointments in case of bacterial infection;-Raising the head higher during sleep in order to reduce morning eyelid swelling.

### 7.3. Treatment of Mild GO

In most cases, GO is a mild, self-limiting disease which undergoes spontaneous remission. Therapeutic management of mild GO is limited to watchful observation and conservative treatment, mainly with the aforementioned topical agents [1,58]. Moreover, in mild GO the following measures can be applied:-The use of 100 μg of selenium twice a day for 6 months in active mild GO of recent onset was effective in re-ducing eye symptoms and improving quality of life, which is attributed to the anti-inflammatory and antioxi-dant properties of this element. Moreover, a long-term therapeutic effect after discontinuation of the treatment and a lower incidence of progression to more severe forms of GO were observed [59]. However, it should be emphasized that these data were obtained from a study performed in marginally selenium-deficient areas of Europe. Whether similar beneficial effects of selenium can be observed in GO patients in selenium replete areas has to be investigated.-Subconjunctival botulinum toxin injections to reduce retraction of the upper eyelid (especially effective in pa-tients in the active phase of the disease, when the final surgical correction should not be performed yet) [60].

### 7.4. General Principles of GO Management

In the group of patients with moderate-to-severe GO the severity of the disease has a significant impact on their daily functioning. Therefore, moderate-to-severe GO is an indication for immunosuppressive therapy in the active inflammatory phase or for surgical treatment in the inactive end stage [53]. Patients with moderate-to-severe GO should be referred to specialist centers experienced in the management of GO as well as potential side effects of the applied treatment.

#### 7.4.1. Active Moderate-to-Severe GO-Glucocorticotherapy—First Line Treatment

The therapeutic effect of GCs in the treatment of GO has been widely evaluated in many clinical trials. GCs seem to be effective in the treatment of inflammatory lesions of orbital soft tissues and extraocular muscles, however, according to the literature, their impact on the reduction of exophthalmos is limited. In multiple studies, local and oral GCs administration proved less effective than intravenous therapy, which, according to the current EUGOGO guidelines, is the treatment of choice for active moderate-to-severe GO [61,62,63]. Compared to intravenous GCs, oral therapy was characterized by a worse therapeutic outcome, a higher rate of recurrence of ocular symptoms, a longer treatment period, and a greater percentage of adverse effects typical of iatrogenic Cushing’s syndrome [64]. Periocular injections of triamcinolone (40 mg/mL) proved to have a positive impact on reducing double vision and eye muscles oedema, while subconjunctival administration was shown to decrease retraction of the upper eyelid. Transient increase of intraocular pressure was the only one observed side effect of local administration of triamcinolone [52]. It should be emphasized, however, that precise evaluation of adverse effects associated with the use of intravenous GCs is limited due to the lack of one consistent therapeutic algorithm. The EUGOGO protocol recommends 500 mg of intravenous methylprednisolone (MP) administered at weekly intervals for 6 weeks, then 250 mg every week for the next 6 weeks, with a total cumulative dose of 4.5 g [53]. This regimen proved more effective and had a greater safety profile compared to other regimens evaluated in clinical trials [65]. Higher dose protocols should be reserved for more severe cases of GO, however single dose of MP should not exceed 0,75 g, the total cumulative dose of MP should not exceed 8 g in one treatment cycle and consecutive-day therapy should be avoided [1,53]. The use of oral prednisone between consecutive steroid cycles or after withdrawal of the therapy does not increase the efficacy of the treatment [64].

Nevertheless, it should be emphasized once again that patients with fibrous lesions in the inactive phase of GO do not benefit from the systemic steroid therapy, which should not be used in this group of patients [66]. It should also be noted that before administering GCs, each patient should be assessed for [67]:-Diabetes—metabolically uncontrolled diabetes is a contraindication to GCs treatment,-Uncontrolled resistant arterial hypertension, severe arrhythmias, unstable ischemic heart disease, severe heart failure are contraindications to GCs treatment,-Liver function disorders—a 4–5-fold increase in the activity of liver enzymes is a contraindication to the treatment; moreover, it is necessary to exclude viral hepatitis, and according to some authors, also autoim-mune hepatitis before introducing systemic GCs therapy,-Glaucoma—a complete ophthalmological examination, including measurement of IOP should be performed both before and after a treatment cycle, because GCs increase IOP. However, it should be noted that an in-crease in IOP in GO may also result from the underlying disease,-Infection markers (blood count, CRP and urinalysis). GCs treatment should be postponed in the case of bacteri-al, viral or fungal infections,-Osteoporosis—it is recommended to perform densitometry before long-term (>3 months) treatment with GCs. Moreover, the EUGOGO suggests supplementation with vitamin D3, calcium and the use of anti-resorptive drugs during systemic steroid therapy, especially in patients with multiple risk factors for osteoporosis-Peptic ulcer disease—especially important among patients on chronic non-steroidal anti-inflammatory drugs (NSAIDs). Proton pump inhibitors are recommended during GCs treatment.-Mental disorders—according to some authors, severe mental illnesses are a contraindication to GCs therapy.

The adverse effects of intravenous glucocorticoid therapy that have been reported in the literature include: hypertension; hyperglycemias (requiring insulin in patients without previous carbohydrate metabolism disorders, as well as deterioration of glycemic control in patients with previously diagnosed diabetes); arrhythmias (it is worth noting that slow intravenous infusion of GCs reduces the risk of arrhythmia); acute coronary syndromes; cerebral venous thrombosis; acute liver injury; psychoses and infections. Most of the reported adverse effects occurred during treatment regimens with cumulative doses exceeding 8 g MP or with daily doses greater than 500 mg over a few consecutive days [1].

Partial or not sufficient response to GCs may be an indication for a second course of systemic glicocorticotherapy, if the treatment is well tolerated. However, watchful observation or second line treatment methods (as presented in Figure 2) may also be implemented and the therapeutic decisions should be made individually [53].

#### 7.4.2. Active Moderate-to-Severe GO—Second Line Treatment

##### Radiotherapy on the Orbital Cavities

Radiotherapy (RT) is widely known for its immunomodulatory actions. According to the literature, low doses of RT reveal an immunosuppressive effect mainly by reducing the adhesion of leukocytes to the endothelium, promoting the apoptosis of immune cells involved in inflammatory process, increasing the expression of anti-inflammatory cytokines and decreasing the secretion of pro-inflammatory cytokines, including TNF-α, IL-1β as well as nitric oxide (NO) and reactive oxygen species (ROS) [68,69].

Radiotherapy is commonly used in the treatment of the active form of GO [68]. However, it should be emphasized that, according to many authors, its efficacy is controversial. In vitro studies have shown that radiotherapy inhibits the activity of orbital lymphocytes and fibroblasts, thereby reducing the production of pro-inflammatory cytokines and glycosaminoglycans [70]. Numerous studies have assessed the use of radiotherapy, both as monotherapy and in combination with systemic oral or intravenous steroid therapy. One study revealed that retrobulbar RT (total dose 20 Gy) was equally effective as a 3-month oral prednisone therapy [71]. In another studies, it has been shown that the efficacy and the durability of the therapeutic effects are greater for combined treatment than radiotherapy or GCs alone [72]. Observations show that combined treatment was associated with lower total dose and shorter duration of systemic corticotherapy and with a lower number of relapses of GO [72]. Therefore, this treatment regimen is now standard care protocol in active moderate-to-severe orbitopathy in many Polish centers [73]. However, it should be emphasized that reliable comparison of the efficacy of radiotherapy is limited due to the use of heterogeneous radiation protocols and different clinical response criteria in the available trials. Studies reveal that radiotherapy proved most beneficial for patients with diplopia resulting from the inflammation of the oculomotor muscles [71,73]. So far, no consistent guidelines have been developed that would clearly define indications for radiotherapy in GO and standardize the therapeutic protocol. According to the available literature, the most commonly used and well-tolerated radiation dose is 20 Gy (ten doses of 2 Gy per each eye), although a good clinical response to total dose of 10 Gy in different fractions has also been reported [74].

Orbital radiotherapy may cause transient exacerbation of the ocular symptoms, which, according to the literature, can be limited by the parallel use of low doses of oral GCs [62,64]. The retina, the lens and the lacrimal gland are the eye structures mostly exposed to the side effects of RT. Persistent xerophthalmia, which refers to the chronic sensation of dry eye, is the most common toxicity, which occurs in approximately 12% of patients [75]. In contrast, de novo occurrence of cataracts or retinopathy is rare, however RT may increase the risk of progression of pre-existing retinopathy and is therefore contraindicated in patients with uncontrolled diabetes and hypertension [76]. In addition, due to the potential risk of secondary carcinogenesis, radiotherapy should be avoided in patients below 35 years of age [36,76].

Conventional orbital radiotherapy area is determined basing on the three-dimensional conformal planning. The treatment radiation is delivered in straight lines and all the structures inside the designated area, even the healthy tissues adherent to the target, receive the same amount of radiation. New radiation techniques, for example volumetric modulated arc therapy (VMAT), enable more precise and accurate planning of the radiation area, reducing the radiation of the surrounding healthy tissues, therefore decreasing the risk of the aforementioned side effects [75].

##### Cyclosporine

Cyclosporine is a calcineurin inhibitor that impedes the secretion of IL-2 from T lymphocytes [77]. The efficacy of cyclosporine in monotherapy in GO has not been confirmed, however, combined therapy with cyclosporine and oral prednisone in moderate-to-severe GO was characterized by higher efficacy and fewer relapses than either treatment alone in two randomized placebo controlled studies [78,79]. In the study of Kahaly et al., the treatment protocol was 5 mg/kg of cyclosporine daily given for 12 months with 100 mg of oral prednisone daily in decreasing doses for 3 months, while in the work of Prummel et al. the combined therapy was applied to patients, whose response to monotherapy was insufficient, with doses of cyclosporine of 7.5 mg/kg and oral prednisone of 60 mg daily. The most common adverse effects of the treatment were dose-related hepato- and nephrotoxicity, gingival enlargement and increased blood pressure.

##### Rituximab (RTX)

Rituximab is a monoclonal antibody against the CD20 protein present on the surface of immune B cells. Its potential efficacy in GO is most likely associated with the inhibition of antigen presentation by B lymphocytes, what inhibits the production of autoantibodies and pro-inflammatory cytokines [80]. In a randomized, double-blind study Salvi et al. confirmed the preliminary reports of a better therapeutic effect of RTX (applied at dose 1000 mg given once a week for two weeks, total dose 2000 mg) in active moderate-to-severe GO compared to intravenous MP. According to the authors, the efficacy of RTX in improving the eyeball motility, subjective assessment of patients’ quality of life and decreasing the need for adjuvant surgical treatment indicates that RTX is a disease modifying drug in GO. On the other hand, in a prospective, randomized, double-blind, placebo-controlled study Marius et al. did not observe RTX superiority over placebo in patients with active moderate-to-severe GO of long duration [81,82,83]. According to the literature, there are some reports of DON occurrence after RTX, therefore it should not be used in patients at high risk of DON development. In view of the presented data further large and adequately designed studies have to be undertaken to assess the use of RTX both as a first and second-line treatment of GO. Moreover, investigations to evaluate the relationship between the duration of the disease and the efficacy of RTX therapy as well as the association between RTX and DON should be performed.

#### 7.4.3. Alternative Treatments with Potential Efficacy

Intravenous GCs remain the treatment of first choice in active moderate-to-severe GO. However, the presence of steroid resistance, steroid dependence or significant adverse effects of systemic GCs therapy have contributed to an intense search for agents, which, due to their mechanism of action, may be effective in the treatment of GO. However, there are no adequately designed, randomized, multicenter clinical trials covering a sufficiently large group of patients that would assess clinically relevant endpoints. This limits the possibility of comparing the efficacy of these therapies.

##### Somatostatin Analogues (SSAs)

It was confirmed that somatostatin receptors are present on the surface of cells involved in GO pathogenesis, including orbital fibroblasts [84]. The use of a long-acting SSA—octreotide—was effective in reducing inflammation of the orbital soft tissues, but the positive effect on the oculomotor muscles was significantly lower than after GCs therapy [85]. On this basis, it was suggested that octreotide may be a therapeutic alternative in the case of steroid intolerance. Lanreotide, another first-generation long-acting SSA, also reduced the inflammatory process of the orbital soft tissues in active GO. Moreover, the lack of relapse in the final evaluation indicated a long-term therapeutic effect of this therapy [86]. Furthermore, pasireotide, a second-generation SSA, proved to be as effective as intravenous MP in reducing clinical symptoms of active moderate-to-severe GO [87]. As of yet, the high cost of treatment and the lack of randomized, prospective trials involving sufficient group of patients have limited the use of SSAs in the treatment of GO.

##### Azathioprine

A multicenter, double-blind, randomized, controlled trial carried out by Rajendram et al. showed that combined use of azathioprine and oral prednisolone was characterized by a better and long-term clinical response compared to GCs monotherapy. The authors suggested that combination therapy may reduce the risk of GO relapse after withdrawal of steroid therapy [88].

##### Mycophenolate Mofetil (MMF)

The mechanism of action of mycophenolic acid is based on selective inhibition of DNA replication in T and B lymphocytes [89]. According to the available literature, the use of MMF in active moderate-to-severe GO was associated with a better therapeutic effect than GCs, especially with regard to the improvement of diplopia and the reduction of exophthalmos [90]. Furthermore, Kahaly et al. in their multicenter randomized trial on a group of 164 patients with active moderate-to-severe GO compared the efficacy of MP monotherapy and combination therapy with MMF, showing the superiority of the combination therapy [91].

##### Methotrexate (MTX)

Methotrexate—a folic acid antagonist, a chemotherapeutic, an immunosuppressive drug—reduces the activation of T and B lymphocytes [92]. In 2019, Yong et al. published a retrospective review of clinical data on 72 patients with severe GO and compressive optic neuropathy treated with MP as monotherapy or in combination with methotrexate. The combination therapy proved more effective than MP monotherapy. However, it should be emphasized that the use of combination therapy was not associated with reduction of the total dose of GCs [93].

##### Tocilizumab (TCZ)

Tocilizumab is a human monoclonal antibody directed against the IL-6 receptor, which is one of the cytokines involved in the immune process underlying GO [94]. In a prospective, non-randomized interventional study by Pérez-Moreiras et al. on a group of patients with active GO resistant to intravenous steroid therapy TCZ decreased activity of the disease (defined by lowering the CAS), reduced exophthalmos and improved ocular motility. The authors did not observe any significant side effects of the therapy or relapse of GO after treatment withdrawal, therefore suggesting there is a possibility of using TCZ in case of steroid resistance [95].

##### TNF-Alpha Inhibitors

Etanercept, a TNF-alpha inhibitor [96], reduced the CAS in patients with GO and severe symptoms of active inflammation [81,83,95,97]. However, larger, randomized, controlled trials are needed to further evaluate the effect of anti-TNF treatment and compare its side effects with those associated with GCs therapy.

##### Teprotumumab

As described above, IGF1 receptors are involved in enhancing the signaling pathway induced by TRAb ligation to TSH-R. The blockage of IGF1-R inhibits the stimulatory effect of TRAb antibodies on orbital fibroblasts.

Recently, intense investigations on Teprotumumab, an IGF1-R antagonist, were performed. The results of a multicenter randomized double-blind placebo controlled trial with 86 patients with active moderate to severe GO, in which half of the patients were receiving Teprotumumab (10 mg/kg for the first infusion and 20 mg/kg thereafter once every 3 weeks for 21 weeks) and the other half placebo, are spectacular [98]. The primary endpoint, defined as reduction of proptosis by at least 2 mm was achieved in 83% of patients treated with Teprotumumab versus 10% of those who received placebo. The number needed to treat was 1.36, indicating that almost every patient benefited from the therapy. The mean reduction of proptosis for the treatment group was 3 mm. It should be emphasized that, so far, such effects were observed only after surgical interventions. These effects were the resultant of the reduction of both extraocular muscles size and retrobulbar tissues. Secondary endpoints, including CAS reduction, improvement of diplopia and of the quality of life were also more favorable in the treatment group comparing to placebo. A significant clinical improvement was observed after the first 6 weeks of the therapy. The tolerance of Teprotumumab was satisfactory, with only mild side effects including hyperglycemia, hearing impairment and muscle cramps. Basing on the aforementioned results on the 21 of January 2020 Teprotumumab was approved by the Food and Drug Administration (FDA) in the treatment of active moderate to severe GO in adults [99]. As all of the immunosuppressants and biological medicaments evaluated so far have been used off-label, it should be noted that Teprotumumab is the first officially registered biological drug in GO treatment. Thus, whether there will be a complete change in the treatment of GO remains an open speculation. This is potentially possible in the future, because the beneficial effect of Teprotumumab is proven, however, many clinical questions require to be answered soon. First of all, the safety profile of the molecule is so far not fully assessed, especially comparing to intravenous GCs, where most of the side effects are commonly known. Furthermore, the long-term treatment effects are still undefined—whether they are permanent or not. Moreover, the issue that is raised especially by European Centers, is the fact that, to date, there are no studies that compare the efficacy of Teprotumumab to intravenous GCs, which currently remain the most effective and first line treatment of GO. Finally, the answer to the question whether Teprotumumab treatment will reduce the need for rehabilitative surgery in GO patients has to be investigated.

#### 7.4.4. Supplementary Treatment

##### Statins

Statins are commonly known as hypolipidemic drugs; however, many studies have also proven their direct pleiotropic anti-inflammatory and immunomodulatory effects [100]. In the literature there are reports of an association between higher serum cholesterol concentration and the occurrence and severity of ocular symptoms in GD [101,102,103]. In addition, administration of statins has been associated with a lower occurrence of orbitopathy in GD, which has not been observed in the case of other hypolipidemic drugs [104,105,106]. It needs to be verified whether the observed relationships result from the anti-inflammatory or lipid-lowering effect of statins. Independently, these observations reveal that statins can potentially be used as an adjuvant drug enhancing the efficacy of immunosuppressive therapy.

##### Enalapril

Enalapril, one of the angiotensin-converting enzyme inhibitors (ACE-I), in addition to its anti-hypertensive properties has also an anti-proliferative effect [107]. In vitro studies on orbital fibroblasts of GO patients and of healthy individuals revealed that enalapril inhibited hyaluronic acid production in both groups [108]. In addition, recent in vivo studies showed improvements in both ocular parameters and the subjective assessment of patients’ quality of life after enalapril treatment [109]. However, there is a lack of large, double-blind, placebo-controlled clinical trials evaluating the in vivo efficacy of enalapril in GO patients.

##### Pentoxifylline

Pentoxifylline, widely used in peripheral vascular diseases, exhibits complex immunomodulatory effects by influencing the signaling pathways and the production of cytokines such as TNF-alpha, IL-1, TGF-alpha [110]. In vitro studies revealed that the use of pentoxifylline reduced the proliferation of orbital fibroblasts derived from patients with active GO, inhibiting their conversion to adipocytes and, consequently, reducing the production of glycosaminoglycans [111]. In a study of Chang et al. the authors assessed the effect of pentoxifylline on the course of inactive GO. A three-month treatment with pentoxifylline administered intravenously at a dose of 200 mg/day for 10 days, then orally 1800 mg/day for 4 weeks, followed by reduction to 1200 mg/day was associated with significant improvement of the quality of life of the patients. Moreover, in a trial of Balazs et al., objective measurements of exophthalmos before and after treatment supported the hypothesis that pentoxifylline could potentially be used in the inactive phase of GO [112]. However, in order to accurately assess the efficacy of pentoxifylline as a potential additive therapy it is still necessary to undertake large, randomized and controlled trials.

#### 7.4.5. Inactive Moderate-to-Severe GO—Surgical Treatment

Although surgical treatment of GO is beyond the scope of this work, it should be emphasized that, except for severe or sight-threatening orbitopathy, surgeries are performed after the active inflammatory process has subsided for at least 6 months [53]. It is recommended that invasive procedures in GO are performed in specialistic clinics experienced in ocular surgery. Surgical treatment is performed pursuant to both medical and cosmetic indications. The most common procedures are orbital decompression, consisting in partial removal of one or more bone walls of the orbit in order to reduce exophthalmos, lower intraocular pressure, improve the venous blood and lymph outflow, decrease soft tissue edema and, as a result, reduce pressure on the optic nerve [113]. For this purpose, also orbital adipose tissue removal is performed [114]. In cases of permanent diplopia, oculomotor muscle surgeries are performed [115]. Surgical eyelid lengthening and blepharoplasty improve the excessive retraction of the eyelids [115]. It needs emphasizing that surgical procedures, if needed, should be performed in the presented above order—beginning with orbital decompression, through squint surgery and concluding with eyelid surgery, as each proceeding step may influence another one [53].

### 7.5. Treatment of Sight-Threatening GO

In approximately 3–5% of cases the course of GO is severe and may lead to complete loss of vision. In clinical practice, this is most commonly associated with optic neuropathy and, to a lesser extent, with corneal ulceration or globe subluxation [84]. Optic neuropathy may occur when the swelling of the extraocular muscles exerts pressure on the neurovascular bundle at the apex of the orbit. There have also been reports of neuropathy caused by excessive stretching of the optic nerve in severe exophthalmos [84]. Optic neuropathy should always be suspected in cases of significant impairment of visual acuity, of the intensity or quality of the perceived color and when the motility of the eyeball becomes suddenly restricted [1]. In such cases, immediate implementation of the treatment is required with intravenous steroid therapy being the treatment of choice. The most common protocol in case of DON is MP in a dose of 500 mg to 1 g for 3 consecutive days. This regimen may be repeated the following week, however, in case of insufficient clinical response or significant adverse effects, immediate orbital decompression surgery should be performed [113]. The superiority of decompression surgery as first-line therapy over systemic steroid therapy has not been demonstrated in active sight- threatening orbitopathy [116].

## 8. Conclusions

Graves’ orbitopathy significantly reduces patients’ quality of life. For this reason, the efficacy of many potential therapeutic options has been widely assessed in the recent years. Management of GO is difficult and requires interdisciplinary cooperation in endocrinology, ophthalmology, radiation oncology and surgery. Currently, intravenous MP therapy is the treatment of choice for active moderate-to-severe GO. The growing knowledge on the pathogenesis of the disease has contributed to the use of immunosuppressive drugs, in example cyclosporine, as a second line treatment. Other immunosuppressants, like azathioprine, mycophenolate mofetil or methotrexate, may be potentially effective in the treatment of GO, however, the possibilities of undertaking a reliable assessment and comparison of the efficacy of the aforementioned drugs are limited due to the heterogeneity of the available studies conducted on small groups of patients. Monoclonal antibodies and anti-cytokine drugs have been widely assessed in multiple clinical trials, however, high cost of such therapies significantly limits its use in clinical practice. Further studies should be undertaken to assess the efficacy of these new therapeutic options and to compare them with classic systemic steroid therapy. The registration by FDA of Teprotumumab, an IGF1-R antagonist, in January 2020 may be a milestone in future management of active GO. However, many clinical questions require to be investigated first, including the efficacy of Teprotumumab in comparison with intravenous GCs, long-term treatment effects, safety profile, and whether this treatment would reduce the need for rehabilitative surgery.

## Figures and Tables

**Figure 1 jcm-10-00016-f001:**
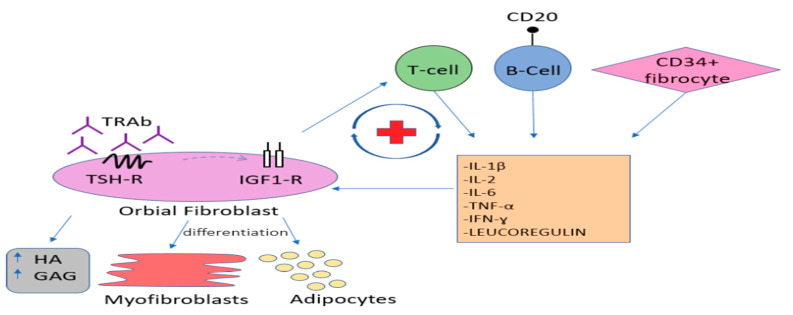
Signaling cascade involved in Graves’ orbitopathy (GO) pathogenesis.IGF1-R: insulin-like growth factor 1 receptor, TSH-R: thyrotropin receptor, TRAb: Thyrotropin receptor antibodies, HA: Hyaluronic Acid, GAG: glycosaminoglycans, IL-1β: interleukin-1β, IL-2: interleukin-2, IL-6: interleukin-6, TNF-α: tumour necrosis factor alpha, IFN-y: interferon gamma.

**Figure 2 jcm-10-00016-f002:**
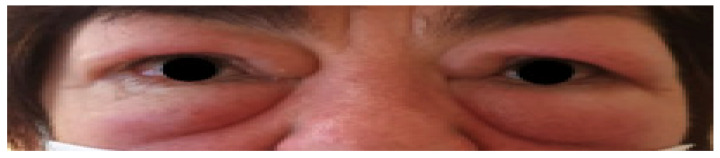
Presents a clinical picture of a 55 years old female admitted to our clinic with active GO (Graves’ Orbitopathy). Medical history of 131-I treatment of hyperthyroidism in the course of GD (Graves’ Disease) two years before.

**Figure 3 jcm-10-00016-f003:**
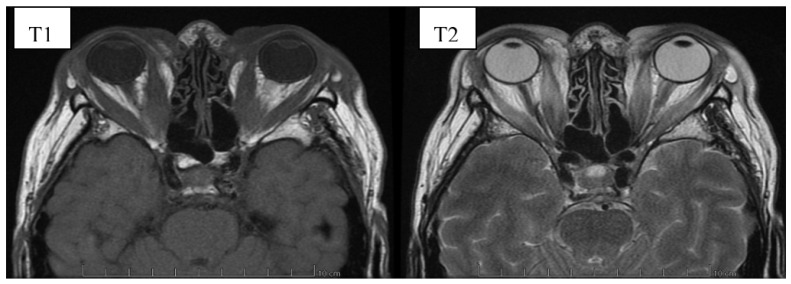
Orbital magnetic resonance imaging (MRI) of a female patient, whose clinical picture was shown on Figure 2. The examination was performed without intravenous administration of paramagnetic. We can observe significantly widened outlines of the straight muscles, the outlines of the extraocular muscles with increased signal intensity in T2-dependent images, ocular adipose tissue with features of edema. Also, edema present in the tissues of both eyelids.

**Figure 4 jcm-10-00016-f004:**
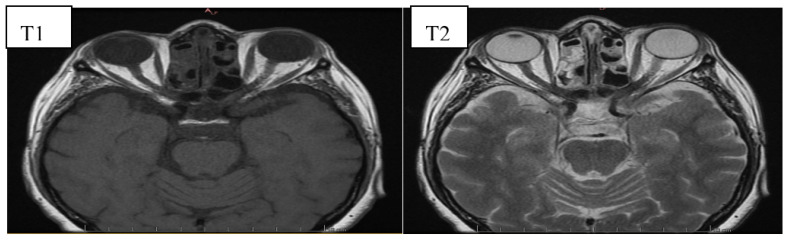
Transverse magnetic resonance imaging (MRI) image of the orbits of a 60 years old female patient with active GO, in whom the orbital changes were asymmetrical—the left orbit was more involved. The examination was performed in T1 and T2 sequences. The left inferior rectus muscle is significantly edematous, the left medial and lateral rectus muscle and the right inferior rectus muscle are also slightly edematous— the picture indicates active inflammatory orbital changes in the course of the underlying disease. Moreover, there is exophthalmos of the left eyeball and a slightly increased amount of the fluid in the left optic nerve sheath compared to the right side. There are no deviations of the lacrimal glands or the extra-conial adipose tissue.

**Figure 5 jcm-10-00016-f005:**
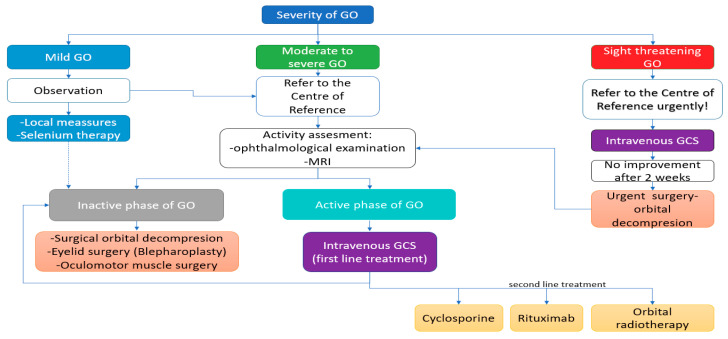
The management of GO (Graves’ Orbitopathy) depending on the disease severity and activity according to EUGOGO (The European Group on Graves’ Orbitopathy) guidelines [53]. MRI: magnetic resonance imaging, GCS: Glucocorticoids.

**Table 1 jcm-10-00016-t001:** This Risk factors for Graves’ orbitopathy.

**Gender**	- In women GO is more frequent [26,27]. - In men GO is more severe [26].
**Race**	- In Caucasians GO is more frequent than in Asians [29].
**Genetics**	- Mostly similar to Graves’ disease [23]. - Some studies evaluated immunomodulatory genes including: human leukocyte antigen-DR3 *(HLA-DR3)*, interleukin-1 *(IL-1)*, IL-23 receptor *(IL-23R)*, *CD40*, cytotoxic T lymphocyte antigen *(CTLA-4),* T-cell receptor β-chain *(TCR-β)*, protein tyrosine phosphatase non-receptor type 22 *(PTPN22)*, tumor necrosis factor-β *(TNF-β)* and numerous immunoglobulin heavy chain-associated genes [25]. - Due to the involvement of TSH-R into the patoghenesis of GO the TSH-R gene polymorphisms were studied [24]. However, none of the polymorphisms have proved adequately predictive to support genetic testing in determining prevention methods and further diagnostic and therapeutic process. - Moreover, the increased orbital adipogenesis in GO contributed to genetic testing of the adipogenesis-related gene peroxisome proliferator–associated receptor-γ *(PPAR-γ)* [24].
**GD duration**	- The longer duration of GD-related hyperthyroidism the higher risk of GO [30].
**Age**	- Older age of GD onset is associated with higher risk of GO development [30]. - Older age of GO onset is associated with more severe course of the disease [26].
**Exogenous factors**	Active or passive smoking: Higher risk of de novo development of GO [14,24] More severe course of GO [33,34,35] The outcomes of the GCs treatment are delayed and limited [33,34,35] Higher TRAb titers and longer persistence during/after the treatment of GO [31] Higher de novo occurrence or exacerbation of GO after radioiodine in smokers [36,37] Radioactive iodine therapy [5,31].
**Biochemical factors**	- Thyroid dysfunction—both hyper and hypothyroidism—is associated with a greater risk of development, progression, and severe course of GO compared to euthyroid patients [38]. - High TSHR antibody titers increase the risk of GO development, positively correlate with the activity and severity of the disease and are a predictor of poor response to the to immunosuppressive treatment and the risk of relapse after treatment [31].

GO: Graves’ orbitopathy.

**Table 2 jcm-10-00016-t002:** Clinical Activity Score—CAS (amended by EUGOGO after Mourits et al.). One point is given for the presence of each of the parameters assessed, the sum defines clinical activity [9].

Initial CAS Assessment, Scored by Points 1–7	
Pain	1 Spontaneous orbital pain 2. Gaze evoked orbital pain
Redness	3. Eyelid erythema 4. Conjunctival redness that is considered due to active GO
Swelling	5. Eyelid swelling 6. Chemosis 7. Inflammation of the caruncle or plica
**Follow-Up CAS Assessment (after 1–3 Months Period), Scored by Points 1–10**	
Impaired function	8. Decrease of eye movements in any direction above ≥5° (during a 1–3 months period) 9. Decrease of visual acuity of ≥1 line on the Snellen chart (during a 1–3 months period)
Proptosis	10. Increase of proptosis ≥2 mm (during 1–3 months period)

**Table 3 jcm-10-00016-t003:** This is a table. NO-SPECS ((No signs or symptoms; Only signs or symptoms; Soft tissue involvement; Proptosis; Extraocular muscle involvement; Corneal Involvement; Sight loss) classification [41].

Class	Abbreviation	Description	Detailed Description
O	N	No signs or symptoms	No complaints, No findings in physical examination (PE)
1	O	Only signs, no symptoms	No complains, PE: Eyelid retraction Stare
2	S	Soft tissue involvement	Swelling of eyelids Chemosis Photophobia Grittiness
3	P	Proptosis	Exophtalmus
4	E	Extraocular muscle involvement	Restricted eyeball mobility (often diplopia)
5	C	Corneal involvement	Keratitis, Corneal Ulcer
6	S	Sight loss	Decreased visual acuity, impaired color of vision (optic nerve involvement)

**Table 4 jcm-10-00016-t004:** VISA (Vision, Inflammation, Strabismus and Appearance) classification. Patients with moderate inflammatory index (less than 4 of 10) are managed conservatively. Patients with high scores (above 5 of 10) or with evidence of progression of the inflammatory process are offered a more aggressive therapy [42].

Sign/Symptom	Score
Caruncular edema	0: Absent 1: Present
Chemosis	0: Absent 1: Conjunctiva lies behind the grey line of the lid 2: Conjunctiva extends anterior to the grey line of the lid
Conjunctival redness	0: Absent 1: Present
Lid redness	0: Absent 1: Present
Lid edema	0: Absent 1: Present but without redundant tissues 2: Present and causing bulging in the palpebral skin, including lower lid festoon
Retrobulbar ache: -At rest -With gaze	0: Absent 1: Present
	0: Absent 1: Present
Diurnal variation	0: Absent 1: Present

**Table 5 jcm-10-00016-t005:** EUGOGO protocol to assess the severity of Graves’ ophthalmopathy. Some of the signs may be assessed by comparison with the image atlas provided by the EUGOGO [40].

Soft Tissues	Eyelid swelling 1. Absent 2. Mild: none of the features defining moderate or severe swelling are present 3. Moderate: definite swelling but no lower eyelid festoons and in the upper eyelid the skin fold becomes angled on a 45° downgaze 4. Severe: lower eyelid festoons or upper lid fold remains rounded on a 45° downgaze
	Eyelid erythema 1. Absent 2. Present
	Conjunctival redness 1. Absent 2. Mild: equivocal or minimal redness 3. Moderate: <50% of definite conjunctival redness 4. Severe: >50% of definite conjunctival redness
	Conjunctival edema 1. Absent 2. Present: separation of conjunctiva from sclera present in >1/3 of the total height of the palpebral aperture or conjunctiva prolapsing anterior to grey line of eyelid
	Inflammation of caruncle or plica semilunaris 1. Absent 2. Present: plica is prolapsed through closed eyelids or caruncle and/or plica are inflamed
Eyelid Measurements	Palpebral aperture (mm)
	Upper/lower lid retraction (mm)
	Levator function (mm)
	Lagophthalmos 1. Absent 2. Present
	Bell’s phenomenon 1. Absent 2. Present
Proptosis	Measurement with Hertel’s exophthalmometer. Recording of intercanthal distance.
Ocular Motility	Prism cover test Monocular ductions Head posture Torsion Field of binocular single vision
Cornea	Corneal integrity 1. Normal 2. Punctate keratopathy 3. Ulcer 4. Perforation
Optic Neuropathy	1. Visual acuity (Logmar or Snellen) 2. Afferent pupil defect (present/absent) 3. Color vision 4. Optic disc assessment: normal/atrophy/edema

EUGOGO: The European Group on Graves’ Orbitopathy.

**Table 6 jcm-10-00016-t006:** This is a table. Severity classification of GO. The management of patients depends on the severity which is established according to the impact of the disease on the patient’s quality of life and the risk of vision loss. The disease is classified as mild, moderate, severe, or sight-threatening as follows [1].

Mild	GO has a minor impact on the patient’s everyday life. They usually present one or more of the following signs: 1. Minor lid retraction (<2 mm) 2. Mild soft tissue involvement 3. Exophthalmos < 3 mm (above the normal range for the race and gender) 4. Transient or no diplopia 5. Corneal exposure with a good response to lubricants.
Moderate to severe	patients without sight-threatening GO whose eye disease has sufficient impact on daily life to justify the risks of immunosuppression (if active) or surgical intervention (if inactive). Patients usually present one or more of the following signs: 1. Lid retraction (>2 mm) 2. Moderate or severe soft tissue involvement 3. Exophthalmos ≥3 mm (above the normal range for the race and gender) 4. Inconstant, or constant diplopia.
Sight-threatening GO	Patients with dysthyroid optic neuropathy or corneal breakdown due to severe exposure. Other infrequent cases are ocular globe subluxation, severe forms of frozen eye, choroidal folds, and postural visual darkening. This category warrants immediate intervention.

GO: Graves’ Orbitopathy.

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
