# Peer review of "Current Knowledge on Graves’ Orbitopathy"

_jcm, 2020, doi:10.3390/jcm10010016_

Round 1

Reviewer 1 Report

This review is about Graves' orbitopathy. The manuscript has a logical flow and structure and it is quite well written.

Broad generalisations based on other review articles rather than original research characterise this paper and there were not so much new information compared with prior review articles in Graves' orbitopathy.

In order to make this paper more suitable and helpful to the medical readers, I suggest an extensive revision of manuscript

My suggestions are:

-focused and reviewed one aspect of GO and draw out all the related open questions in order to identify areas of specific medical need that should be investigated in future research (future prospective)
-add author’s opinion and practical advice “when” o “who” manage the GO patients would be helpful to the reader and make the paper more useful

Reviewer 2 Report

Dear authors,

thank you for an interesting literature review concerning Graves' orbitopathy. Management of GO requires interdisciplinary cooperation in endocrinology, ophthalmology, radiation oncologist and surgery. Importantly, due to limited data with small patient cohorts therapeutic strategies remain controversial.

The manuscript is well written. However, some improvements are required:

Chapter: 2.1. Clinical picture and diagnosis

Please include if possible a clinical image and cranial MRI (image)

Please include a chapter and a table concerning reliable risk factors (please include several references)

Please elaborate the chapter of radiotherapy in more detail (see e.g. https://www.ncbi.nlm.nih.gov/pmc/articles/PMC6010590/ )

Please include a chapter about imaging and treatment response evaluation (e.g. https://pubmed.ncbi.nlm.nih.gov/31792849/ )

Reviewer 3 Report

In the proposed manuscript the authors present the current status of knowledge about Graves' orbitopathy (GO).

the manuscript is mostly well written and pretty comprehensive, therefore I only have minor points:

1) I think the "etiopathogenesis" part would benefit from the addition of a figure that schematize the signaling cascades discussed in the paragraph.

2) paragraph 3.2 (treatment of mild GO): although it is true that selenium has been reported to reduce symptoms in mild GO the paper cited by the authors (ref 43) shows data obtained in areas of Europe that are known to be selenium deficient. therefore the authors should mention the existence of such controversy.

Round 2

Reviewer 1 Report

Congratulation!No other comments for authors

Reviewer 2 Report

The manuscript has been significantly improved. It is well written and pretty comprehensive, therefore I have nothing to contribute. Congratulation